# On Synthetic Data and Iterative Magnitude Pruning: A Linear Mode Connectivity Study

## Abstract

Recent works have shown that distilled data representations can be leveraged for accelerating the training of DNNs. However, to date, very little is understood about the effect of these synthetic data representations in the area of architectural optimization, specifically with Iterative Magnitude Pruning (IMP) and pruning at initialization. We push the boundaries of pruning with distilled data, matching the performance of traditional IMP on ResNet-18 & CIFAR-10 while using 150x less training points to find a sparsity mask. In our novel approach, we find that distilled data guides IMP to discard parameters contributing to the sharpness of the loss landscape, fostering smoother landscapes. These synthetic subnetworks are stable to SGD noise at initialization in settings when the dense model or subnetworks found with standard IMP are not, such as ResNet-10 on ImageNet-10. In other words, training from initialization across different shuffling of data will result in linear mode connectivity, a phenomenon which rarely happens without some pretraining or weight permutation. This inherent stability proves to be useful for finding lottery tickets in larger settings, where traditional IMP cannot. Since synthetic subnetworks are constructed at initialization, these can be used as the stable starting point for future IMP steps, finding tickets at significantly higher sparsities than IMP without pretraining. This behavior is heavily linked to the compressed representation of the data, highlighting the importance of synthetic data in neural architectural validation. In order to find both a high performing and robust sparse architecture, a more optimal synthetic data representation is needed that can compress irrelevant noise like distilled data, yet better maintain task-specific information from the real data as dataset complexity increases.

## 1 Introduction

Sparse neural networks are increasingly important in deep learning to enhance hardware performance (e.g., memory footprint, inference time) and reduce environmental impacts (e.g., energy consumption), especially as state-of-the-art foundational models continue to grow significantly in parameter count. In parallel, researchers have been exploring how synthetic data representations such as those generated by dataset distillation methods can be leveraged to efficiently accelerate deep learning model training. With this in mind, we explore the training dynamics and stability of sparse neural networks in the context of synthetic data to better understand how we should be efficiently creating sparsity masks at initialization. Modern pruning methods only reduce computational costs at inference. When pruning at initialization, we can greatly reduce training costs as well.

Research on the training dynamics of dense models have led researchers to find that dense models are connected in the loss landscape through nonlinear paths (Freeman & Bruna, 2017; Draxler et al., 2019; Garipov et al., 2018). Linear paths or Linear Mode Connectivity (LMC) is an uncommon phenomena that only occurs in rare cases, such as MLPs on subsets of MNIST in Nagarajan & Kolter (2021) or model intervention with weight permutation (Ainsworth et al., 2023). For large networks, Frankle et al. (2020) found that pretrained dense models when fine-tuned across different shufflings of data, are linear mode connected. While these models are "stable" to noise generated through stochastic gradient descent (SGD), only the smallest dense models are stable at initialization. As for its relationship with sparse neural networks, Frankle et al. (2020) empirically found that the Lottery Ticket Hypothesis only holds for stable dense models, those that are linear mode connected across

data shuffling. They found that these large dense models only become stable early in training, leading to the conclusion that Iterative Magnitude Pruning (IMP) with weight rewinding, the method to find such lottery tickets, should instead rewind a model to an early point in training rather than at initialization. We support these conclusions as we find that both dense and IMP-generated subnetworks are not stable at initialization. Our work aims to study sparse neural networks at initialization, different than new age lottery ticket literature (Paul et al., 2022), which utilizes some pretraining to find a good "initialization". Finding high performing sparsity masks at creation is the ultimate goal of neural network pruning for both efficient training and gaining fundamental understanding of sparse neural networks.

We find that another class of sparse subnetworks exist that are more stable at initialization: *synthetic subnetworks*. We define synthetic subnetworks as those produced during pruning with distilled data. These are found by replacing the traditional data in IMP with distilled data, essentially a summarized version of the training data consisting of only 1-50 synthetic images per class (see Sachdeva & McAuley (2023) for a survey). We perform the same training, pruning, and rewinding to produce the sparsity mask. This mask, as with those produced by IMP, can be applied to the dense model at initialization to create a high performing sparse neural network after training on real data. The significance is that synthetic images can be used to pick an appropriate sparsity mask for a downstream task. These subnetworks have a lower need for rewinding to an early point in training due to their inherent stability. We find that, with dataset distillation methods such as Kim et al. (2022), we match performance with IMP when rewinding to initialization. We achieve this with approximately *150x less* training points than traditional IMP to find a sparsity mask. While we do use a current state-of-the-art distillation method, such methods are still limited to small datasets like CIFAR-10, CIFAR-100, and ImageNet-10. The representation of this data is purely responsible for this performance. We explore more in depth how the structural differences in these sparsity masks provide more stability than IMP-generated masks. We further investigate why synthetic subnetworks are stable when the dense model is not by generating loss landscapes visualizations and study what these characteristics could mean for the future of sparse neural network research. Finally, we apply this inherent stability to find lottery tickets in settings larger than previously discovered.

## 2 PRELIMINARIES AND RELATED WORK

### 2.1 SPARSITY IN NEURAL NETWORKS

The most common form of sparsity can be found in neural network pruning literature (Gale et al., 2019). In this field, researchers exploit sparsity for computational savings, usually at inference. Pruning can be divided into two categories based on the parameters being removed: Structured and Unstructured. Unstructured has no regulation on what can be pruned, the typical example is removing individual weights. By pruning weights, the weight matrices are still the same size, but contain zeroed out weights. Structured pruning is removing parameters that make up some physical pattern, such as removing neurons, channels, attention heads, etc. All of these examples physically make the weight matrices smaller, since you can remove a whole row or column at once. These lead to faster inference on general hardware; however, the added restrictions reduce the performance of the pruned model in comparison to unstructured methods.

*When* to prune is an entirely other consideration: before training (Lee et al., 2019), during training, or after training (Han et al., 2015). In order to reduce the cost of training as well, Lee et al. (2019); Tanaka et al. (2020) explore how to prune at initialization. This is essentially producing optimal sparse architectures for downstream tasks, the end goal for almost any pruning research. Despite these great ambitions, pruning at initialization does not perform as we hope (Frankle et al., 2021). To further understand why this is the case, Frankle & Carbin (2019) propose the Lottery Ticket Hypothesis: for a sufficiently over-parameterized dense network, there exists a non-trivial sparse subnetwork that can train in isolation to the full performance of the dense model. They employ iterative magnitude pruning with weight rewinding to retroactively discover these sparse subnetworks, named "Lottery Tickets" due to the lucky nature of winning the initialization lottery. They empirically proved that pruning at initialization is indeed possible in smaller settings.

## 2.2 STABILITY TO STOCHASTIC GRADIENT DESCENT

To uncover why the early iteration of the Lottery Ticket Hypothesis did not apply to large networks, Frankle et al. (2020) finds that this hypothesis only holds when the dense model is "stable" to the effects of data shuffling. At initialization, only the smallest computer vision models on MNIST are stable to SGD noise without weight permutations (Ainsworth et al., 2023; Entezari et al., 2022). A model is deemed stable if training across different orders of training data result in models with linear mode connectivity. Using this knowledge, iterative magnitude pruning can be adapted to instead rewind to weights back to an early point in training, a point in which the dense model is stable. This previous work empirically proves that we can prune at some early point in training for larger models. In our work, we intentionally rewind to initialization rather than early in training in order to learn how to prune at initialization. Rewinding to early training only empirically proves the existence of subnetworks with matching performance during early training, but it does not accurately reflect how sparse networks behave from random initialization.

## 2.3 DATASET DISTILLATION

We explore the idea of synthetic data in pruning by using current methods under dataset distillation. In general, dataset distillation optimizes a synthetic dataset to match the performance of a model trained on real data. This bi-level optimization problem can be defined as minimizing the difference of average loss over all validation points:

$$\arg\min_{\mathcal{D}_{\text{syn}}} |L(\Phi(\mathcal{D}_{\text{real}}); \mathcal{D}_{\text{val}}) - L(\Phi(\mathcal{D}_{\text{syn}}); \mathcal{D}_{\text{val}})| \tag{1}$$

or, the minimizing the supremum of the absolute loss difference across over each validation pair as proposed in Sachdeva & McAuley (2023), a unifying survey work:

$$\arg\min_{\mathcal{D}_{\text{syn}}} \left( \sup\{ |L(\Phi(\mathcal{D}_{\text{real}}); (x, y)) - L(\Phi(\mathcal{D}_{\text{syn}}); (x, y))| \}_{(x,y)\sim\mathcal{D}_{\text{val}}} \right) \tag{2}$$

where $\Phi$ is a training algorithm returning optimal parameters for the dataset $\mathcal{D}_{\text{syn}}$. Many methods include matching the training trajectories of a student and teacher model (Cazenavette et al., 2022), meta model matching (Wang et al., 2020), or factoring (Deng & Russakovsky, 2022). A simplification of this process for classification tasks is compressing each set of images for a label down into a few images. While this view of minimizing the performance difference of real data and synthetic data is constrictive as it never allows for synthetic data to perform better than real data, we show that it is a reasonable starting point from which we use to explore the training and stability dynamics of synthetic data representations.

## 2.4 CHALLENGES WITH DISTILLED DATA

Current dataset distillation methods do not properly maintain relevant information on the downstream task for harder datasets, but they do sufficiently minimize the image-specific information resulting in over-compression. These issues arise as a result of the high computational complexity of optimizing a synthetic training set. Current methods cannot maintain performance as the dataset or model complexity increases, limited to convolutional networks on toy datasets like CIFAR and MNIST. For example, we use Efficient Dataset Condensation (Kim et al., 2022) in this paper which achieves 50.6, 67.5, or 74.5% accuracy on CIFAR-10 depending the the size of the synthetic dataset (1,10, or 50 images per class or ipc) or 72.8 & 76.6% accuracy on ImageNet-10 (Russakovsky et al., 2015). While the aim of dataset distillation literature is to have a small synthetic set to improve training efficiency on other architectures, less aggressive compression may be needed. We suggest that efficiency is not the only use case or goal with distilled data. Compressing the training set may result in better generalization, as long as relevant information is maintained.

# 3 DATASET DISTILLATION FOR NEURAL NETWORK PRUNING.

We employ Iterative Magnitude Pruning with Distilled Data for the first time to discover the architectural biases that this data brings. Specifically, we use dataset distillation methods that match training trajectories to ensure that training on synthetic data yields similar converged results to training on real data (Kim et al., 2022). We refer to this algorithm as *Distilled Pruning*.

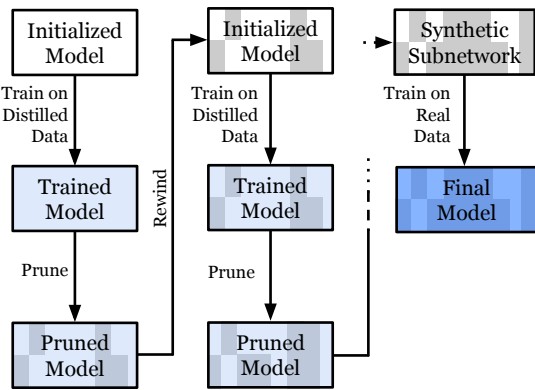

Figure 1: Distilled Pruning Algorithm Diagram

As shown in Figure 1, to find a suitable sparsity mask of a randomly initialized model, we first train the network to convergence on distilled data, prune the lowest magnitude weights, then rewind the non-pruned weights back to their initialized values, and loop until desired sparsity. The final model should have its randomly initialize weights with a sparsity mask. We can train the sparse synthetic subnetwork on real data to achieve sufficient performance at high sparsities. Using distilled data only to choose our sparsity mask allows us to better understand the biases that this data introduces in the architecture. We refer to subnetworks found with synthetic or distilled data as *synthetic subnetworks* and those with real data as *IMP subnetworks*.

The only differences of IMP and distilled pruning lie in the sparsity mask they choose. Since each method uses different datasets for training, their final converged weights will be different. Weights deemed "important" for the real dataset may not be important on the distilled data; distilled pruning may attempt to remove these. This is also true for the other direction. The performance of these sparsity masks by distilled pruning directly relates to how relevant the distilled data is to the real data. Figure 2 showcases the performance of this method compared to traditional IMP on CIFAR-10 with ResNet-18.

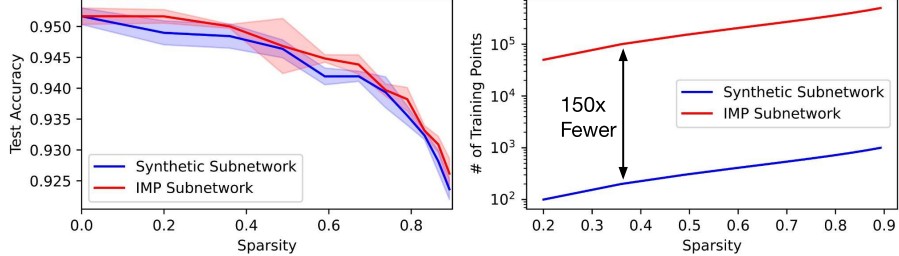

Figure 2: Performance of Distilled Pruning vs Traditional IMP on ResNet-18 & CIFAR-10. The distilled dataset consisted of 10 images per class. Error bars are plotted as we average across 4 seeds. The plot on the right measures the amount of data points used in training to find a sparsity mask at x sparsity. Note that in IMP we are not finding "lottery tickets" since we rewind back to initialization for both methods, not to an early point in training.

We see in Figure 2 that lower test accuracy after training on distilled data does not completely translate in neural network pruning. Despite only achieving 72.8% after training on distilled CIFAR-10 with 10 ipc from Kim et al. (2022), we can use this low performance to choose weights to prune

matching the performance on IMP on smaller datasets. To be clear, distilled data is only used to find a sparsity mask, the sparse model is ultimately trained on real data for validation. As previously mentioned, this performance does not hold as dataset complexity increases, we find that CIFAR-100 can perform decently at lower sparsities, but does not handle extreme sparsities ($> 90\%$) with larger models like ResNet-18. In those cases, the distilled data on CIFAR-100 is not maintaining task-relevant information for the model. Since distilled data contains less outliers, and is a largely more "generalizable" dataset, we find that the sparsity mask pruned weights that control the fine grained details of the real data.

## 4 STABILITY OF SUBNETWORKS.

To understand the training dynamics of sparsity masks chosen via distilled pruning vs IMP, we conduct an instability analysis. We take a randomly initialized model, generate a sparsity mask through pruning, and train it across two different orderings of the real training data. We save these two models and interpolate all the weights between them, measuring the training loss at each point in the interpolation as shown in Figures 3 and 4. We assess the linear mode connectivity of these subnetworks to determine if the model is stable to SGD noise. If the loss increases as you interpolate between two trained versions, then there is a barrier in the loss landscape, implying the trained models found different minima. In these cases, the ordering of the training data directly impacts what minima its choosing. If the loss does not increase during interpolation, then this implies they exist in the same minima or at least the same flat basin.

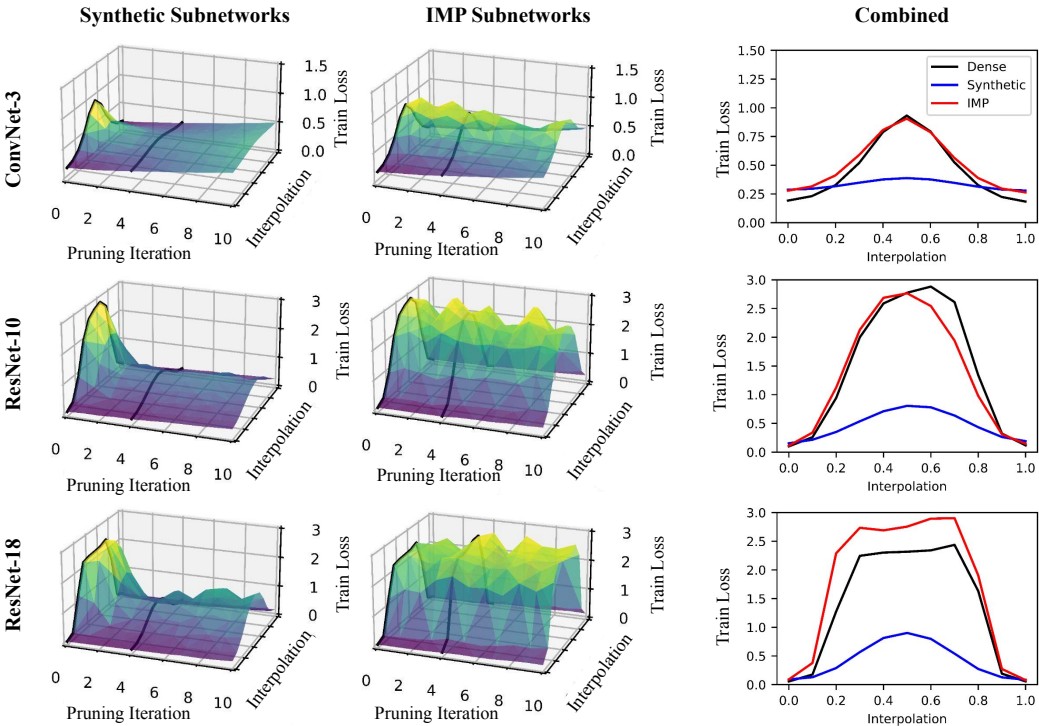

Figure 3: Comparison of the stability of synthetic vs. IMP subnetworks at initialization on CIFAR-10. We show how the loss increases as you interpolate the weights between two trained models. We measure this for subnetworks of different sparsities. The left column is reserved for subnetworks found via distilled data, and the middle column is for subnetworks found with real data. We aggregate all the information in each row for a better comparison. The dark lines in the 3D plots represents the pruning iteration we used for the combined plot; the dense model is iteration 0.

We see that in simpler scenarios with ConvNet-3 on CIFAR-10 & ResNet-10 on ImageNet-10, we exhibit full linear mode connectivity. We even see slightly better performance during interpolation in Figure 4. We find that in some cases of unstable dense models there exists a sparse subnetwork that is

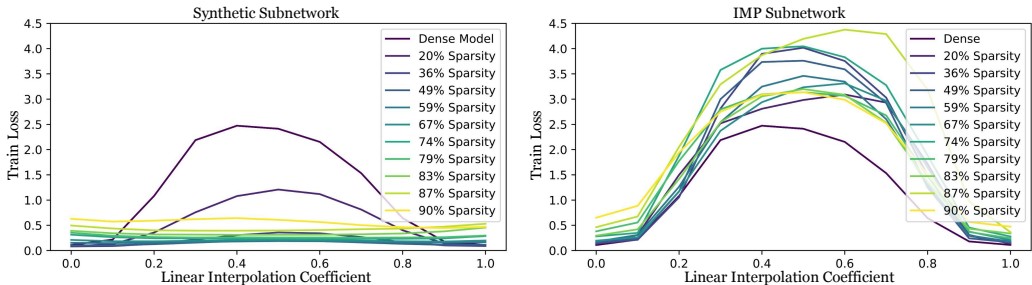

Figure 4: Comparison of the stability of synthetic vs. IMP subnetworks at initialization on ImageNet-10 and ResNet-10. An increased loss across interpolation implies instability / trained networks landing in different minima.

stable at initialization. More importantly, traditional IMP is not able to produce stable subnetworks in these settings. Sparsity is not necessarily the answer for smoother landscapes, *where* you induce sparsity is the main factor. As pruning continues, the results exhibit more stability despite lower trainability, as seen with higher training losses. We postulate that the parameters pruned on distilled data, yet still exist in the IMP subnetwork, capture the intricacies of the real data which contribute to a sharper, but more trainable, landscape. Since IMP subnetworks are not stable, the intricacies it is learning is order dependent.

## 4.1 Loss Landscape Visualization

While linear mode connectivity is useful to study the loss landscape, this lightweight method can only show us a one dimensional slice of the bigger picture. We further examine the landscapes across two dimensions of parameters as shown in Figure 5.

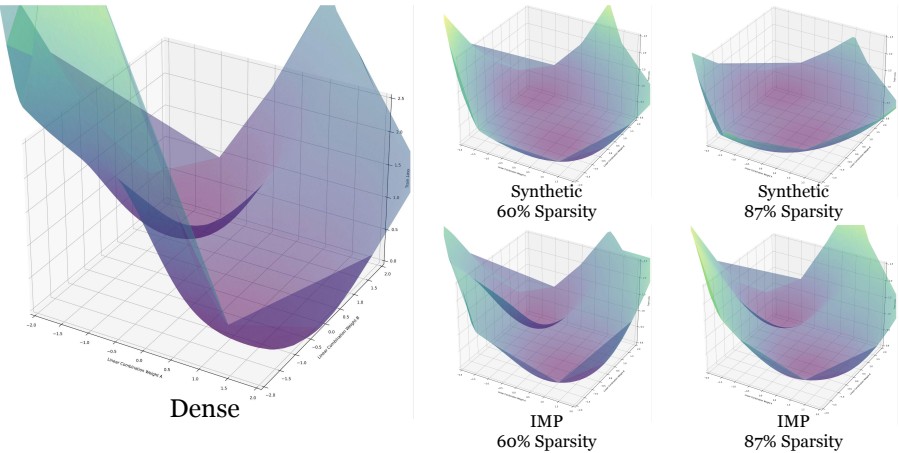

Figure 5: Loss Landscape visualization around the neigbhorhood defined by trained models on different seeds for ConvNet-3 and CIFAR-10.

We trained the same network across 4 different seeds, much like in linear interpolation; however, we also created 2 orthogonal vectors from this to sample nearby points. This maps the hyperdimensional parameter space down to two dimensions, since every point shown in our visualization is a linear combination of the two vectors. For each of the 10,000 points, we recorded the loss validated by real training data. Since this visualization is created after reference models are trained, reference models that are closer together will result in "zooming in" on their minima. The spatial distance is not preserved using this method. This is useful in determining the local area in which these models are training to. With post-hoc analysis, we find that spatial distance in our plot is mainly maintained, with slightly lower distances as you prune.

From these visualizations, IMP chooses subnetworks that exhibit a similar landscape to the dense model. We see the trained models fall into two separate minima in both the IMP and Dense cases, explaining the loss barrier in the Figure 3. Subnetworks chosen with distilled data are falling into the same, flat basin. We also validated these visualizations quantitatively by looking at the diagonal of the hessian. We find that synthetic subnetworks have a tighter distribution of diagonal hessian values around zero than both the dense model and IMP subnetworks. Though, sparsity in general leads to small decreases in sharpness which can also be seen in the Figure 5 visualization [1].

## 4.2 Hessian Analysis

For a more quantitative approach to measuring sharpness of the loss landscape, we measure the diagonal of the hessian for a batch as computing the full hessian is computationally prohibitive. We study the diagonal for ConvNet-3 on CIFAR-10.

| Subnetwork | Sparsity | Min/Max | Mean ± Std | Avg Magnitude |
|:---:|:---:|:---:|:---:|:---:|
| Dense | 0 | -1.847 / 1.344 | .0048 ± .035 | .0067 |
| IMP | 60% | -3.073 / 4.510 | .0051 ± .0559 | .0087 |
| IMP | 90% | -6.323 / 6.127 | .0035 ± .0594 | .0060 |
| Synthetic | 60% | -0.982 / 2.093 | .0047 ± .0286 | .0061 |
| Synthetic | 90% | -2.338 / 1.730 | .0037 ± .0354 | .0053 |

We see that across every measure of the distribution of the diagonal hessian, small amounts of pruning in traditional IMP lead to sharper landscapes. This smooths out as we approach higher sparsities though. With our synthetic subnetworks, the distribution is much closer to zero with less variance. We see the same pattern of smoother values as sparsity increases. While smoothness from increased sparsity may be a result of lower dimensionality of the landscape, synthetic subnetworks are still behaving radically different than IMP-chosen subnetworks at the same sparsity.

## 4.3 Overview of Synthetic Subnetworks

We examine how varying the compression rate of dataset distillation affects the performance and stability of synthetic subnetworks. Across datasets, performance of synthetic subnetworks will be compared to its IMP counterpart. Performance is expressed as the *ratio* of synthetic subnetwork accuracy over IMP subnetwork accuracy. We compare stability similarly against the IMP subnetworks. We plot stability as the barrier height ratio between the Synthetic and IMP subnetworks. Barrier height is measured as the distance between the trained loss and the loss of the interpolated model from our linear mode connectivity experiments. Both a low and high sparsity example is reported for each setting. 60% sparsity is chosen as the low sparsity as this tends to be when synthetic subnetworks first achieve general stability. We include a single marker for IMP to showcase what "equivalent" algorithm to IMP looks like, as the ratio of IMP to itself is 1. Note, on ImageNet-10 we exhibit a negative barrier height at high sparsity.

Across almost all experiments, we see a general trend: subnetworks chosen via distilled pruning result in a smooth & generalizing loss landscape. As compression ratio increases, we see more stability than IMP; however, the performance trend largely depends on the distilled accuracy in Kim et al. (2022). Most notably, we achieve full linear mode connectivity for ConvNet-3 on CIFAR-10 and ResNet-10 on Imagenet-10. While there are numerous factors at play, Kim et al. (2022) optimized the synthetic data specifically for these models, hinting that stability is a result of high performing synthetic data. As distillation methods improve, we expect to maintain label-information (pushing the points up) and be able to use fewer images per class (pushing the points to the left).

## 5 Stable Initializations & The Lottery Ticket Hypothesis

Distilled Pruning does not have to compete with IMP; it can be used in tandem with it. We study how these The Lottery Ticket Hypothesis only holds when the initialization is stable. Even on settings as small as ResNet-18 on CIFAR-10, Iterative Magnitude Pruning cannot produce lottery tickets at

---

[1]See the appendix for results on the distribution of diagonal hessian values.

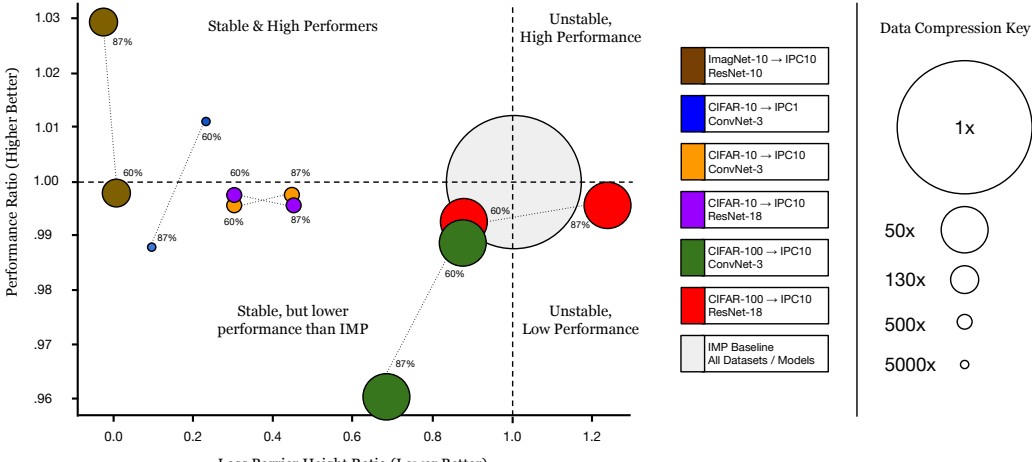

Figure 6: Comparison of synthetic subnetworks and IMP subnetworks across models & datasets. Both the 60% and 87% sparsity model is chosen for each example and connected with a dotted line. These are compared to IMP's 60% and 87% sparsity respectively. The size of the marker represents the amount of data compression through dataset distillation used to find the sparsity mask. For example, distilling CIFAR-10 (default 5,000 images per class) down to 10 images per class is a 500x compression ratio. Performance is measured as the ratio of test accuracy between Synthetic and IMP subnetworks. Loss Barrier Height is measured by subtracting the trained loss and the halfway interpolated trained loss from linear interpolation experiments.

high sparsities as shown in Fig. 2. To get around this, researchers use weight rewinding to some point in early training when the model is stable Frankle et al. (2020). We refer to rewinding point as the $k$th epoch in training. Since these networks are not pruned at initialization, rewinding instead to some $k > 0$, these are not lottery tickets; they are only considered *matching subnetworks*. Since synthetic subnetworks are found by rewinding to initialization, we can use this instead as our stable starting point, no pretraining needed. In combination, distilled pruning removes weights that are responsible for instability, then IMP can choose a high performing subset of stable weights.

Building on our study of ResNet-10 and Imagenet-10, we use a synthetic subnetwork that is both stable and high performing as our sparse initialization, then perform traditional iterative magnitude pruning. Following this criteria, we chose to prune ResNet-10 over 8 iterations or 83% sparsity with distilled pruning, then performed IMP for 25 iterations. Figure 7 provides a comparison between this, traditional IMP with rewinding back to initialization ($k$th epoch is 0), and IMP rewinding to the 3rd epoch.

We demonstrate that using synthetic subnetworks as a stable starting point can produce lottery tickets at higher sparsities than previously thought. Figure 7 also shows that it is possible to regain performance lost from the synthetic initialization. The results for rewinding to early in training, $k = 3$, were previously considered to be a practical upper bound for pruning at initialization.

## 6 DISCUSSION

Synthetic sparsity masks are a result of pruning irrelevant parameters on an already compressed dataset. Weights appearing in the IMP subnetworks, but not synthetic, are used to capture the fine-grained details of the real dataset. The removal of these parameters leads to smoother landscapes; however, this can lead to worse performance, especially on distilled datasets that are not aligned well to the training set such as CIFAR-100. Upon analysis of these sparsity masks in isolation, we could not draw any conclusions about sparsity patterns. Distilled Pruning leads to extremely similar sparsity per layer, with a seemingly independent distribution of pruned parameters. We invite future researchers to engage in this analysis. The only differences we observed with these models is how they learn on real data, such as stability & smoother loss landscapes.

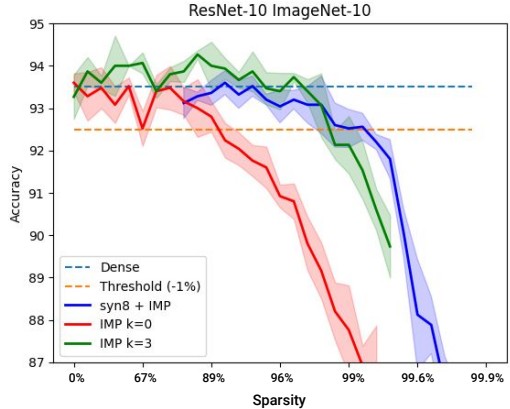

Figure 7: Performance comparison of traditional IMP ($k = 0$), IMP with weight rewinding to the 3rd epoch in training ($k = 3$), and traditional IMP performed on a stable, synthetic subnetwork at 83% sparsity. We observe synthetic initializations find lottery tickets at higher sparsity than traditional IMP ($k = 0$) alone. We emphasize $k = 3$ is not finding lottery tickets as this is not at initialization. Each trial ran for 25 iterations with 5 seeds each.

This work is an initial step into exploring the impact of using synthetic data, specifically distilled data, to improve pruning. We thoroughly assess the linear mode connectivity of these subnetworks to determine if the model is stable to SGD noise. Since this work shows how the data used to prune directly impacts the generalizability and robustness of these subnetworks, we aim to rethink the use of any data in these scenarios, not just synthetic. Potentially more aggressive data augmentations or even generative approaches can be used only during pruning to create certain effects in the sparse architecture. We reopen the idea of finding lottery tickets at initialization on larger models by focusing on pruning parameters leading to instability first. We believe that a new algorithm, not the naive combination of distilled pruning and IMP, will be essential for uncovering the Lottery Ticket Hypothesis and validating it on larger settings.

## REPRODUCIBILITY STATEMENT

We aim to make advanced neural network pruning methods more accessible to a broader range of researchers and developers, reducing the expertise and compute overhead required to prune high-performing networks. We provide the source code and configurations for the key experiments including instructions on how to use the distilled data, prune the models with distilled pruning, prune with IMP, and training and validation procedures. Our code is currently available at an anonymous URL: https://anonymous.4open.science/r/SyntheticDataForIMP-D0C8/. The data distillation method we use can be found at https://github.com/snu-mllab/Efficient-Dataset-Condensation. We used the pre-computed distilled data from the original paper (Kim et al., 2022). We use the same implementation of ConvNet-3 and ResNet-10 (with batch normalization) as the dataset distillation method.

## ETHICS STATEMENT

Sparse neural networks offer a practical way to reduce the computational overhead for training and inference which directly translates to lower $CO_2$ emissions and contributes to more sustainable AI research and development practices. Likewise, our paper highlights that leveraging distilled data during the neural network pruning process can result in significant computational savings. However, one possible risk of this approach is the loss of interpretability when using synthetic data (distilled or generated). Specifically, if a practitioner does not have access to the original source data the synthetic dataset was derived from, it may be challenging to understand the limitations, outliers, and biases of that synthetic dataset. To counter the potential risk of unexpected or biased model performance, it is crucial to define frameworks that thoroughly validate that the models trained on synthetic data adequately represent the associated real-world problem domains and tasks. While synthetic data representations hold considerable promise for the future enhancement of deep learning, the drawbacks, nuances, and related mitigation strategies should continue to be carefully studied.

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
