# OpenReview forum: "On Synthetic Data and Iterative Magnitude Pruning: a Linear Mode Connectivity Study"
_ICLR.cc/2024/Conference — Submitted to ICLR 2024_

### Official Review · Reviewer_6HLj · 2023-10-25

**Soundness:** 3 good
**Presentation:** 1 poor
**Contribution:** 2 fair
**Rating:** 5
**Confidence:** 3

**Summary:**

This paper studies the effects of pruning on distilled datasets (the synthetic data created by dataset distillation) and compares it with Iterative Magnitude Pruning (IMP). The authors find that pruning on distilled datasets has a higher efficiency, causes flatter landscapes, and has better linear mode connectivity. They also interpret the phenomena via the information bottleneck perspective.

**Strengths:**

* The information bottleneck perspective on the distilled dataset is interesting and novel.
* The findings that the pruned networks on the distilled dataset have flatter landscapes and better linear mode connectivity are new and insightful.
* The experiments are somewhat solid.

**Weaknesses:**

* **Presentation and structure.** I think this paper may have poor presentation and language. I recommend the authors to polish the paper and reorganize the structure. For example, the information bottleneck part seems strange to me at first glance. Maybe the authors can elaborate more on how information bottleneck is related to the main findings and claims and conduct a more fluent transition between (sub)sections.
* **Novelty.** Using the distilled dataset for better efficiency in pruning has already been proposed in previous literature [1], which weakens the novelty and contribution of the proposed method. Therefore, the efficiency cannot be a main claim. And I think the whole novelty of this paper is weak.
* **Lack of further evidence.** Information bottleneck is a good perspective to understand the distilled datasets, but I think the paper lacks further evidence on how information bottleneck is related to the findings. Concretely, how the information bottleneck is quantized? I think the loss landscape is not a direct aspect to show the point. More direct and intuitive evidence is needed.
* **Lack of implications to practices.** Knowing the fact that pruning on distilled data is not new in this paper, the author should provide more insights on how the findings can guide applications and practices.

------
[1] McDermott L, Cummings D. Distilled Pruning: Using Synthetic Data to Win the Lottery[J]. arXiv preprint arXiv:2307.03364, 2023.

**Questions:**

* The used models are only ResNets and ConvNets. I am interested in the results regarding more model architectures, specifically, Transformer is of particular interest, and other architectures, such as MLPs, VGGs, and MobileNets, are also needed.

---

> ### Author Response · Authors · 2023-11-22
> **Response to reviewer 6HLj**
>
> Weaknesses on Presentation / Structure: I agree. I have revised my language and removed all connection to the IB framework. I will push this idea to future work, as it deservers extensive experiments to back up this connection.
>
> Weakeness on Novelty:
>
> I made a mistake introducing the paper “Distilled Pruning: Using Synthetic Data to Win the Lottery” by McDermott & Cummings as this work is a nonarchival extended abstract, not a full paper. I now have removed all reference of this paper in my submission, and I hope this submission will be treated as if this algorithm, or even the idea to use distilled data with pruning, is introduced for the very first time. I request that you do not look into the authors identity for further reviewing. Hopefully, this will improve the novelty of this paper, as there are no “published” works that explore this idea.
>
> Weaknesses on further evidence:
> See above, I agree and have removed this.
>
> Weaknesses on lack of implications:
>
> To address the concerns for practical implications, I have added a new section that discusses what you can do with these synthetic subnetworks. I have shown with experiments that you can use synthetic subnetworks as a stable initialization for IMP, finding Lottery Tickets at higher sparsity. Previously, I only mentioned this in the body of the paper. I originally did not include this information in the paper as I aimed to focus more on the why the stability is happening, drawing parallels to the IB framework. It appears this paper has been reviewed less as a study and more as pruning algorithm work, so this new section seems more fitting.
>
> Questions:
> I completely agree that we need to see this on transformer architectures; however, because dataset distillation is so new, they cannot yet extend to compute-intensive transformer models. For the others, yes, there is always some additional architectures / datasets to include.
>
> Thank you for your feedback, I appreciate your concerns and have adjusted the paper accordingly. For novelty, I am not at liberty to go into the specifics; however, I please ask that you give an additional rating that assumes this work is novel and has never been published before. I realize after these reviews, I should not have included the original extended abstract on Distilled Pruning. I hope without the novelty concerns, your rating will significantly increase. I have also provided a separate section to address your points of lack of implications, see section 5.

---

### Official Review · Reviewer_rRcD · 2023-10-27

**Soundness:** 2 fair
**Presentation:** 1 poor
**Contribution:** 1 poor
**Rating:** 3
**Confidence:** 5

**Summary:**

The paper investigates the effect of using distilled datasets for training deep neural networks in terms of the linear mode connectivity of pruned models (analogously to experiments by Frankle et al. with iterative pruning). Empirical investigation shows that sparsity masks found with distilled (synthetic) data are approximately as good as the ones found with standard IMP, but at the same moment they demonstrate more stability to SGD noise in terms of linear mode connectivity, i.e., the barrier between "synthetic" subnetworks is smaller than one between standard IMP subnetworks.

**Strengths:**

The idea to understand the effect of the synthetic data on the training loss surface is interesting. It can shed light on the ways data affects the optimization landscape. It also has a benefit of a faster pruning process, since training on less data is (hopefully) faster.

**Weaknesses:**

While the core idea of the investigation is interesting, the overall contribution of the paper is questionable. The largest part of the paper is dedicated to the description of the process of data distillation, while that is not the contribution of the paper (existing distilled datasets are used for the experiments).

The discussion about information bottleneck is largely misplaced in this work. The initial idea of IB is that a deep learning model implicitly tries to learn a representation that keeps maximal possible information about targets and minimal possible information about inputs, thus forming a bottleneck. It is still not proven that such compression is needed for generalization and that deep neural networks indeed perform it. Moreover, a natural conjecture about using a data that is already optimized in terms of information is that the model (neural network in this case) does not have to learn it anymore. Meaning, that from the initialization its task is simplified significantly, it basically does not have to form a bottleneck in itself. The empirical evidence in this paper is supporting this conjecture directly (especially with the observation that for more complex datasets there is no effect of smoothing barriers). I see the analysis of direct differences between sparsity masks induced by distilled data and natural data and analysis of them as an important experiment not performed in this paper. As well as the exhaustive comparison between the performance of synthetic masks should be one of the central and most discussed results, but it is reduced to one diagram in the paper.

Minor:

- please make use of \citep and \citet to distinguish citations that are not part of the sentence and part of it correspondingly

- in section4 there is a mention of "architectural relationship of the data" - it is completely unclear what does this term mean

- diagram in Fig.7 is very hard to understand. Why only 60 and 87 sparsity are chosen? Why the barrier is always same for IMP setup? What is performance ratio? Why IMP baseline is only one no matter that there are several setups?

- I think the first sentence of Discussion does not belong to the text

- the flatness investigation is left out to the appendix, nevertheless it is mentioned as one of the conclusions for the paper. There is no clear connection in general in the existing research between LMC and flatness, so the conclusions are inaccurate.

**Questions:**

1 - What is the core goal of the research performed in the paper?

2 - How easy it is to produce distilled datasets analogous to ones used for experiments? Why they are tightly bound to a particular architecture?

---

> ### Author Response · Authors · 2023-11-22
>
> Disclaimer: I made a mistake introducing the paper “Distilled Pruning: Using Synthetic Data to Win the Lottery” by McDermott & Cummings as this work is a nonarchival extended abstract, not a full paper. I now have removed all reference of this paper in my submission, and I hope this submission will be treated as if this algorithm, or even the idea to use distilled data with pruning, is introduced for the very first time. I request that you do not look into the authors identity for further reviewing. Hopefully, this will improve the novelty of this paper, as there are no “published” works that explore this idea.
>
> I have completely removed this section and any reference to the IB framework in this paper. I agree in retrospect, this seems out of place and would require a full paper with experimentations to draw this connection.
>
> I have attempted to include such an analysis on sparsity masks; however, I have not been able to find such pattern in weight connectivity / sparsity mask. Even considering hamming distance over time, the sparsity masks look indistinguishable. The only differences I am able to conclude is how these sparsity masks perform or train on the real data. I have now added this information in the discussion.
>
> I hoped for this work to be seen as a study of the linear mode connectivity in these models, not a paper pushing the performance of the algorithm. However, to acknowledge Reviewer 6HLj’s concern for practical implications, I have added a new section that discusses what you can do with these synthetic subnetworks. I have shown that you can use synthetic subnetworks as a stable initialization for IMP, finding Lottery Tickets at higher sparsity. I originally did not include this information in the paper as I aimed to focus more on the why the stability is happening, drawing parallels to the IB framework. It appears this paper has been reviewed less as a study and more as pruning algorithm work, so this section seems more fitting.
> Minor:
> - I have instances between \citep and \citet
> - Changed this line.
> - 60% and 87% are two values chosen to mimic “Medium Sparsity” and “High Sparsity”. 60% sparsity refers to Pruning Iteration 4 in Figure 4. Which is where nearly all the “effects of distilled data” on stability begin. 87%, or high sparsity, is chosen when both IMP and Distilled Pruning start to drop off in performance.
> The barrier is always the same because this figure represents the performance $\textbf{ratio}$ distilled pruning compared to IMP: $Acc_{\text{Syn}} / Acc_{\text{IMP}}$.  The performance ofIMP divided by itself is 1, regardless of setting. I have added a few things in the Overview of Synthetic Subnetworks section to make it simpler, but overall this information was included.
> - Removed IB framework entirely.
> - I have now included this in the main body of the paper.
>
> I disagree, in the extreme if you have a completely flat loss landscape, all points are linearly mode connected. Sharper landscapes, even with the same number of minima/maxima will have large barrier heights, leading to poor stability.
>
> Question #1: Traditional Iterative Magnitude Pruning can only find Lottery Tickets on a stable model. Currently, we can only find such stability for this by training for a few epochs, which is not at initialization. My primary research question was, “If a dense network is unstable at initialization, can we find a stable subnetwork here instead?”
>
> Question #2: Very easy. You can download them yourself from the original dataset distillation method’s repository  https://github.com/snu-mllab/Efficient-Dataset-Condensation. However, for datasets not commonly used in distilled data, you will have to rerun this your self. Distilled Data as a field is so new, so I released this paper to show what you can do with it, hopefully encouraging more research on it.
>
> They are tightly bound to a specific architecture because they are biased towards whatever architecture type is used in the optimization procedure for generating distilled data. Because these methods are intensive with significant retraining and trajectory matching, they can currently only scale the architectures so much. Therefore, anything larger than ResNet-50 is not even considered by most modern dataset distillation work. Even then, we do not use ResNet-50 in this paper as these methods perform very poorly at that scale. I hope I made these challenges and issues with current distilled data methods apparent enough in the background section.
>
> I appreciate your feedback on the content of the paper. Your reviews have definitely strengthened my paper and pointed out flaws that I needed a second set of eyes for. For novelty, I am not at liberty to go into the specifics; however, I please ask that you give an additional rating that assumes this work is novel and has never been published before. I realize after these reviews, I should not have included the original extended abstract on Distilled Pruning.

---

> > ### Comment · Reviewer_rRcD · 2023-11-23
> >
> > I thank the author(s) for the reply. I am a bit confused though, because I was not claiming that the paper is not novel, but I got a long explanation about connection to an extended abstract. Nevertheless, I appreciate the answers to _my_ questions as well.
> >
> > About flatness: when it is evaluated using Hessian characteristics, it reflects the property of the surface only in a very small surrounding of the selected point. This will not have an effect (at least obvious one) on the LMC between two random points.
> >
> > Q2. The fact that I can download them does not mean that it is _easy_. What is the complexity of the creation of such dataset? What is the connection with a particular architecture? How it possibly reduces the complexity of the training? All these are the questions that are interesting for such research, but stayed outside of the scope of the paper.
> >
> > While I still agree that the phenomenon itself (stability with distilled data) is interesting, I do not see this paper to be ready for publication and containing enough analysis. I encourage author(s) to continue work.

---

### Official Review · Reviewer_mHnL · 2023-10-29

**Soundness:** 2 fair
**Presentation:** 2 fair
**Contribution:** 2 fair
**Rating:** 5
**Confidence:** 3

**Summary:**

This paper follows the main idea of McDermott & Cummings (2023) that leveraging the synthetic dataset (specifically, dataset distillation) to find a sparsity mask with IMP. The difference between this paper and McDermott & Cummings (2023) is that this paper applies a better dataset distillation method. The strong part of this paper is that they found the subnetworks at initialization that are already stable to SGD noise, which is surprising.

**Strengths:**

1. Sec 5.'s experiments are interesting to me. The authors found that there exits subnetworks are already stable to SGD noise using distilled data.

**Weaknesses:**

1. The contribution is limited. The main idea of this paper actually follows McDermott & Cummings (2023) but change the dataset distillation method.
2. IB framework seems to be irrelevant to the main logic of this paper. A large part of this paper is trying to explain the connection between the IB framework and dataset distillation method, but I don't see a deep connection between this part and the main flow of this paper. Moreover, the IB framework lacks experiment or theoretical analysis which cannot convince me firmly.
3. The Sec 5.1 seems to be redundant. The LMC experiments can already give a clear conclusion over the stability of subnetworks. I don't see a large advantage to include such fancy visualizations in the main text.
4. Some figures seem to be non-informative (e.g., Fig 1 right part and Fig 2). There is no need to explain the dataset distillation and IMP with both texts and figures. This part can be put into a "background" section but no need to explain in such a detailed manner. The audience can be assumed to be people who are knowledgeable in these fields and the effect of including these figures is to lower the informativeness of this paper.
5. The writing of this paper sometimes makes me lost. There are many "therefore" in this paper, but most times when "therefore" occurs there is no clear causal relationship between the sentences before and after. Some expressions are vague, e.g., the "important" in "...What is deemed "important" for real data might not be important for distilled data; therefore, distilled pruning may attempt to remove these...." (P. 5). Also, some expressions are actually wrong, e.g., "Linear paths or Linear Mode Connectivity (LMC) is an uncommon phenomenon that only occurs in rare cases..." and LMC is not a rare phenomenon but happens with both spawning case and permutation case [cite 1].
[cite 1] Samuel Ainsworth, Jonathan Hayase, and Siddhartha Srinivasa. Git re-basin: Merging models modulo
permutation symmetries.
6. Most important references are missing. Only one paper is referenced in the LMC section.

**Questions:**

No question.

---

> ### Author Response · Authors · 2023-11-22
> **Response to Reviewer  mHnL**
>
> Weaknesses 1)
>
> I made a mistake introducing the paper “Distilled Pruning: Using Synthetic Data to Win the Lottery” by McDermott & Cummings as this work is a nonarchival extended abstract, not a full paper. I now have removed all reference of this paper in my submission, and I hope this submission will be treated as if this algorithm, or even the idea to use distilled data with pruning, is introduced for the very first time. I request that you do not look into the authors identity for further reviewing.
>
> I understand where your original ratings stand; however, would it be possible for you to provide a rating of the revised paper that assumes this idea is being published for the first time. You can make a heavy disclaimer with this.
>
> Weaknesses 2)
>
> I have completely removed this section and any reference to the IB framework in this paper. I agree in retrospect, this seems out of place and would require a full paper with experimentations to draw this connection.
>
> Weaknesses 3)
> Section 5.1 showcases similar results, but across an added dimension which shows new information not capture in the linear interpolation plots. Those plots show nothing about the landscape around the trained models, only giving information that the path between them. I disagree that LMC experiments are clear enough, for evidence see Reviewer qh7B’s questions.
>
> Weaknesses 4)
>
> While I agree about comments in weakness 2, I disagree heavily here. Readers who have not read into dataset distillation can be confused on how we generate these representations in the first place. I have faced numerous questions and confusion in the past all about distilled data, even the point of this paper is only to use it. Figure 2 explains when we are adding distilled data, this is not traditional IMP. In the texts, I stress that we are not using distilled data as the final training stage for evaluation; however, I have received comments believing that we are, or even that the loss landscape figures are for distilled data. The audience is not entirely people who are knowledgable in this field because both fields — linear mode connectivity and dataset distillation — are small. One motivates of this paper is to invite researchers interested in pruning and neural architecture evaluation methods like NAS to look into synthetic data as an option, which is an entirely separate field.
>
> Weaknesses 5) I have fixed wording, I agree with the criticisms of language.
>
> For your criticisms on the commonness of Linear Mode Connectivity, this is purely semantics. Git Re-basin shows that after $\textit{permuting}$ a model, they are linear mode connected. LMC across permutation is entirely out of scope of this paper; however, I have now updated the intro/background to at least acknowledge its existence. In addition, I feel that my section title of “Linear Mode Connectivity” in the background was misleading. Changing this to discuss stability to SGD noise seems more appropriate.
>
> Weaknesses 6) Relating to weakness 5, we need LMC without permutations to actually find Lottery Tickets. I referenced works in the intro that motivate the study of linear mode connectivity such as (Nagarajan & Kolter, 2021), but also discuss the study of nonlinear mode connectivity with  (Freeman & Bruna, 2017), (Draxler et al., 2019), (Garipov et al., 2018). I now have added permutation LMC references to acknowledge this subfield; however, the reader should be able to fully understand the motivation for stability without these.
>
> In addition, to replace the section on IB Framework & to acknowledge Reviewer 6HLj’s concern for practical implications, I have added a new section that discusses what you can do with these synthetic subnetworks. I have shown that you can use synthetic subnetworks as a stable initialization for IMP, finding Lottery Tickets at higher sparsity. I originally did not include this information in the paper as I aimed to focus more on the why the stability is happening, drawing parallels to the IB framework. It appears this paper has been reviewed less as a study and more as pruning algorithm work, so this section seems more fitting.
>
> I appreciate your feedback on the content of the paper, and completely agree with most of the comments in hindsight after writing this paper. For novelty, I am not at liberty to go into the specifics; however, I please ask that you give an additional rating that assumes this work is novel and has never been published before. I realize after these reviews, I should not have included the original extended abstract on Distilled Pruning. I hope with these changes, the adjustment to novelty, and the additional section expanding the LTH, you can update your ratings accordingly.

---

> > ### Comment · Reviewer_mHnL · 2023-11-23
> >
> > Thank you for your feedback, especially the clarification on the novelty part. The idea is interesting and still the experiments in sec 5 are interesting to me, and thus I will raise my rating correspondingly. However, I will still keep the rating negative, as this paper needs further polish and I recommend the author to consider the resubmission of this work.

---

### Official Review · Reviewer_qh7B · 2023-11-01

**Soundness:** 3 good
**Presentation:** 2 fair
**Contribution:** 3 good
**Rating:** 6
**Confidence:** 3

**Summary:**

This paper studied the stability of synthetic subnetworks (the lottery ticket at initialization obtained by IMP after training o distilled data) with linear mode connectivity. In contrast to the usual instability observed in both the dense network and the standard IMP subnetwork, the synthetic subnetwork proves to be remarkably stable to SGD noise at initialization. To gain a better understanding of this phenomenon, the paper proceeds to visualize the loss landscape and quantitatively assess its sharpness through hessian approximation.

**Strengths:**

1. This paper offers a compelling insight into the impact of data distillation on the stability of lottery tickets obtained through the IMP method. This observed stability holds true across various datasets and models, strengthening the credibility of the findings.
2. The paper conducts a comprehensive analysis of the performance and stability of synthetic subnetworks. Furthermore, its impressive visual representations provide valuable insights into the study of stability and the intricacies of the loss landscape.

**Weaknesses:**

The analysis of the loss landscape and its sharpness for dense models, synthetic subnetworks, and IMP subnetworks is quite interesting. However, I have a couple of questions that I hope the author can clarify:

1. The loss landscape of dense models and IMP subnetworks appears to be sharper than that of synthetic subnetworks. Can the sharpness of the loss landscape be used as a criterion for determining the quality of a subnetwork?
2. In the last paragraph of Section 5.1 (on page 8), it is mentioned, 'We see the trained models fall into two separate minima in both the IMP and Dense cases, explaining the loss barrier in Figure 4.' This seems to connect stability with the loss landscape. If I understand correctly, the linear path for stability and loss landscape are different. Is this proper to explain the stability with this loss landscape?

Other tiny issues:

1. The axis legend in Figure 6 is almost unreadable.

**Questions:**

1. How is the stability of the other distilled data other than the one evaluated in the paper?
2. How does IPC impact the stability? Can more studies are provided for ResNet18 on CIFAR10 and CIFAR100?

---

> ### Author Response · Authors · 2023-11-20
> **Response to Reviewer qh7B**
>
> In response to 1) in Weaknesses, smoothness of the loss landscape is not exactly a direct criteria for performance (that would be the exact loss value), but it does tell us about generalization and how the model responds to small perturbations. For example, consider the functions $x^2$ and $5x^2$, they have the same minimum 0 when $x=0$, meaning they will “perform” the same at inference on the set they are optimized for. However, $5x^2$ cannot handle small perturbations to x as well as $x^2$, as it has a larger gradient and thus worse loss for some $5(x+\epsilon)^2$ for some small $\epsilon$. While this is a convex example, we see more drastic issues when nonconvex landscapes are sharp. This matters in when you deploy your models as the distribution shifts away from your training data. Also during training time, smoother landscapes are much easier to optimize for.  In summary, they are not an exact 1-1 comparison for loss, but they are much more robust models. Robustness is extremely important for sparse models.
>
> In response to 2) in Weaknesses, the linear path for stability is drawn by interpolating between two trained models. We visualize the loss landscape around converged models, showcasing that we exist in different minima. If you linearly interpolated between minima in the loss landscape, you would be drawing a line through that loss barrier. The linear interpolation barrier from the plots are directly seen in the loss landscape figure. The landscape is essentially giving us 1 more dimension worth of information.
>
> Tiny Issue 1): Fixing this issue. The axes values are not entirely important to understand the smoothness, but I realize it is important to ensure the reader that these values are held constant.
>
> Question #1: Previous methods on Distilled Pruning used MTT (https://arxiv.org/abs/2203.11932) which is a lower performing method than the one we use. Despite lower performance, MTT also can be used to find stability on small cases of AlexNet / CIFAR-10 as shown in previous nonarchival work. Both of these methods aim to match training trajectories, which means the distilled loss landscape & real loss landscape are closely aligned. Other methods have no such requirement, which I hypothesize may not always find stability; however, I have not tried them.
>
> Question #2
>
> This answer is dependent on a few things / settings. Generally, when the images per class decreases the stability increases. This is why we see stability at all with pruning on distilled data rather than real data. As shown in Figure 7, we see this trend with ConvNet-3 on CIFAR-10. I would have added more points to this plot to support this claim; however, it became too cluttered.
>
> For CIFAR-100, I firmly believe dataset distillation is too young as a field to handle are large number of classes, so I would prefer to wait for the next SOTA distillation method to explore this. For CIFAR-10, we illustrate the linear interpolation plots in Figure 4. In addition, we also show the performance of the distilled pruning algorithm on ResNet-18/CIFAR-10 in Figure 3.
>
> Thank you for your responses and questions. If you have anymore, please feel free to comment.

---

### Meta-Review · Area_Chair_pUBM · 2023-12-02

**Metareview:**

This paper presents a pruning method using synthetic data to find the sparsity mask. The main results of the paper are focused on improved stability (measured using linear mode connectivity as in Franckle et al. 2020) close to initialization, as well as other loss landscape visualization results and Hessian analysis.

**Justification For Why Not Higher Score:**

More experiments can be done:
(1) Comparison between sparsity masks learned by different methods;
(2) Solid comparison between synthetic subnetworks and other pruning methods;
(3) Better and clearer demonstrations of the stability at initialization and comparison to other lottery-ticket-finding algorithms. Currently, the results are not fully fleshed out.

Many reviewers mentioned good suggestions to help improve the writing. The authors should consider submitting a stronger version soon.

Last but not least, the best way for the authors to address the novelty issue is to demonstrate the difference and the improvement clearly.

**Justification For Why Not Lower Score:**

N/A

---

### Decision · Program_Chairs · 2024-01-16

Reject